# Influences of Four Kinds of Surfactants on Biodegradations of Tar-Rich Coal in the Ordos Basin by *Bacillus bicheniformis*

**DOI:** 10.3390/microorganisms11102397

**Published:** 2023-09-26

**Authors:** Wensheng Shen, Xiangrong Liu, Chen Shi, Jie Yang, Shunsheng Zhao, Zaiwen Yang, Dan Wang

**Affiliations:** 1College of Chemistry and Chemical Engineering, Xi’an University of Science and Technology, Xi’an 710054, China; shen.wensheng@foxmail.com (W.S.); shichen202306@163.com (C.S.); 18113079003@stu.xust.edu.cn (J.Y.); shshzhao@163.com (S.Z.); yzwxk@foxmail.com (Z.Y.); 2State Key Laboratory of Green and Low-Carbon Development of Tar-Rich Coal in Western China, Xi’an University of Science and Technology, Xi’an 710021, China; dwang@cqu.edu.cn

**Keywords:** tar-rich coal, *Bacillus licheniformis*, biodegradation, surfactant, enrichment

## Abstract

The biodegradation of tar-rich coal in the Ordos Basin was carried out by *Bacillus licheniformis* (*B. licheniformis*) under actions of four kinds of surfactants, namely, a biological surfactant (Rh), a nonionic surfactant (Triton X-100), an anionic surfactant (LAS), and a cationic surfactant (DTAB). The biodegradation rates under the actions of Triton X-100, LAS, Rh, DTAB, and the control group (without surfactant) were 59.8%, 54.3%, 51.6%, 17.3%, and 43.5%, respectively. The biodegradation mechanism was studied by examining the influences of surfactants on coal samples, bacteria, and degradation products in the degradation process. The results demonstrated that Rh, Triton X-100, and LAS could promote bacterial growth, while DTAB had the opposite effect. Four surfactants all increased the cell surface hydrophobicity (CSH) of *B. licheniformis*, and Triton X-100 demonstrated the most significant promotion of CSH. The order of improvement in microbial cell permeability by surfactants was DTAB > TritonX-100 > LAS > Rh > control group. In the presence of four surfactants, Triton X-100 exhibited the best hydrophilicity improvement for oxidized coal. Overall, among the four surfactants, Triton X-100 ranked first in enhancing the CSH of bacteria and the hydrophilicity of oxidized coal and second in improving microbial cell permeability; thus, Triton X-100 was the most suitable surfactant for promoting *B. licheniformis*’s biodegradation of tar-rich coal. The GC-MS showed that, after the action of Triton X-100, the amount of the identified degradation compounds in the toluene extract of the liquid product decreased by 16 compared to the control group, the amount of dichloromethane extract decreased by 6, and the amount of ethyl acetate extract increased by 6. Simultaneously, the contents of alkanes in the extracts of toluene and dichloromethane decreased, lipids increased, and ethyl acetate extract exhibited little change. The FTIR analysis of the coal sample suggested that, under the action of Triton X-100, compared to oxidized coal, the *Har/H* and *A*(CH_2_)/*A*(CH_3_) of the remaining coal decreased by 0.07 and 1.38, respectively, indicating that Triton X-100 enhanced the degradation of aromatic and aliphatic structures of oxidized coal. Therefore, adding a suitable surfactant can promote the biodegradation of tar-rich coal and enrich its degradation product.

## 1. Introduction

Tar-rich coal is a valuable coal-based oil and gas resource with high tar yield, prevalent in both low-rank and middle-rank coal strata within the Ordos Basin [1]. The rich tar and gas resources of tar-rich coal are mainly extracted using pyrolysis technology. However, there are some disadvantages of pyrolysis technology, such as complex equipment, substantial energy consumption, and high environmental pollution [2]. Consequently, it is necessary to develop green and low-carbon technologies to utilize tar-rich coal [3].

The biodegradation of coal features the characteristics of mild conditions, low energy consumption, and simple equipment and is considered a promising technology for coal conversion [4]. In 1981, Fakoussa [5] reported that microorganisms could degrade coal into liquid products. After over 40 years of development, several generally accepted biodegradation mechanisms of coal include the attacks of enzymes, alkaline chemicals, and surfactants on coal [6,7,8]. Surfactants may increase cell surface hydrophobicity (CSH) and then enhance the adsorption of hydrocarbons via microorganisms [9]. Surfactants also change the permeability of the cell membranes, which helps microorganisms absorb coal molecules and release biological enzymes [10]. Moreover, surfactants can increase the solubility of coal by reducing its surface tension [11].

Among them, anionic surfactants and cationic surfactants are relatively low-cost. Nonionic surfactants promote microbial growth with good stability. The biosurfactants are favorable to environmental protection [12]. Li [10] discovered that Tween 80 enhanced the fluidity of the microbial membrane so as to promote the transmembrane transport of hydrocarbons. Zhao [13] pointed out that rhamnolipids (Rh) significantly increased the CSH of cell membrane and enhanced the biodegradation of phenanthrene by *P. aeruginosa.* Xia [14] revealed that dodecyltrimethylammonium bromide (DTAB) could improve the hydrophilicity of coal samples. Our research team also observed that the straight chain alkyl benzene sulfonate (LAS) and polyethylene glycol octylphenyl ether (Triton X-100) improved the biodegradation of coal by changing the hydrophilicity of the coal surface [15,16]. Currently, studies on coal biodegradation mainly focus on lignite and coal slime, with little attention given to tar-rich coal [17,18,19,20,21,22]. The coalification degree of tar-rich coal is higher than that of lignite, which is more difficult to degrade by microorganisms [23]. Therefore, adding Rh, Triton X-100, LAS, or DTAB may be an effective way to improve the biodegradation of tar-rich coal [24,25,26].

*Bacillus licheniformis* (*B. licheniformis*) has been extensively evaluated for its usage in biodegradation of coal [27,28]. Jiang [29] showed that *Bacillus* sp. *Y7* could degrade more than 36.77% of lignite in 12 days. Akimbekov [30] demonstrated that almost 24% of crude lignite was solubilized within 14 days by *Bacillus* sp. Therefore, in this study, *B. licheniformis* was used to degrade tar-rich coal in the Ordos Basin. The optimal biodegradation conditions were obtained through single-factor experiments and orthogonal experiments. The influences of four surfactants—a biological surfactant (Rh), a nonionic surfactant (Triton X-100), an anionic surfactant (LAS), and a cationic surfactant (DTAB)—on the biodegradation were further studied. The biodegradation mechanism was investigated using the growth curve, CSH, cell permeability of *B. licheniformis,* and the contact angle between the coal surface and surfactant solution. The structures and compositions of biodegradation products were analyzed with FTIR and GC-MS. The degradation experimental process is shown in Figure 1.

## 2. Materials and Methods

### 2.1. Coal Pretreatment

The coal samples were mined from the Wujiata mining area in the Ordos Basin, Inner Mongolia, China. The tar in the coal sample was tested by GB/T 1341-2007, and the yield was 9.9% [31]. After being ground, the coal particles with a diameter of 0.2–0.5 mm were used in this study (raw coal). The raw coal was pretreated via nitric acid oxidation in the following steps. The raw coal was added to 8 mol/L nitric acid at the ratio of solid to liquid 1:2.5 (g/mL) and shaken at room temperature for 24 h, then washed with deionized water to pH > 5.6. The coal samples were filtered and dried before use (oxidized coal).

Proximate and ultimate analysis of raw coal and oxidized coal are shown in Table 1. After nitric acid oxidation treatment, the ash content of raw coal decreased from 9.32% to 3.08%, indicating that nitric acid pretreatment removed the ash. Compared with raw coal, the volatiles increased from 28.52% to 40.04% in oxidized coal, which may be caused by the large molecular structure of raw coal oxidized and fractured by nitric acid, and the volatile small molecular compounds were increased. In addition, the oxygen content in oxidized coal increased, while the sulfur content decreased, which indicated an obvious oxidation effect.

### 2.2. Cultivation of Microorganisms

*B. licheniformis* was obtained from the China Center of Industrial Culture Collection (No. CICC 10092) and cultured on nutritive gravy agar medium (5 g peptone, 5 g NaCl, 3 g beef extract, 15 g agar, and 1 L distilled water). The incubation of bacteria was performed at 30 °C in a shaking incubator at 160 rpm and cultured for 3 generations before the biodegradation experiments.

### 2.3. Surfactants

Rh, Triton X-100, LAS, and DTAB were purchased from Aladdin Reagent Co., Ltd (Shanghai, China). These four surfactants could effectively improve the properties of tar-rich coal or cell membranes, affecting coal biodegradation. The structures of the four surfactants are depicted in Table 2.

### 2.4. Biodegradation Experiment

The oxidized coal was added to 150 mL conical flasks with 20 mL of nutritive gravy agar medium. Then, conical flasks were wrapped with paper and sterilized in a vertical autoclave.

After sterilization, the amplification culture of activated bacteria was inoculated and cultured in a constant-temperature shock incubator. The temperature was set at 30 °C, and the oscillation speed was 160 r/min for 18 d.

All experiments were carried out at least in triplicate, and the average value and standard deviation of the obtained data were calculated.

#### 2.4.1. Measurement of Biodegradation Degree

After biodegradation, the degradation liquid products were centrifuged at 10,000 rpm for 20 min. The A450 values of the supernatant liquid were determined using UV–Vis spectrometry. The residual coal was washed with distilled water and dried to a constant weight at 80 °C.

The extent of coal biodegradation was evaluated using the following two methods:(1)The biodegradation activity was assessed by measuring the changes in the A450 values of the supernatant liquid. The A450 was determined using deionized water as a reference.(2)The biodegradation rate was calculated with Formula (1):
(1)$$\eta =\frac{\left(m_{1}-m_{2}\right)}{m_{1}}\times 100\%$$
where *η* represents the biodegradation rate of the oxidized coal, *m*_1_ is the original weight of oxidized coal (g), and *m*_2_ is the weight of residual oxidized coal (residual coal) (g).

#### 2.4.2. Single-Factor Experiment

The influence of the amount of coal, biodegradation time, and inoculation volume on the biodegradation of coal were investigated. Biodegradation conditions are shown in Table 3. All other conditions remained the same: amount of coal, 0.7 g; biodegradation time, 12 d; inoculation volume, 8 mL.

#### 2.4.3. Orthogonal Experiment

The amount of coal, biodegradation time, and inoculation volume were taken as three factors, and each factor was divided into three levels. Using an *L*9 (3 × 3) orthogonal design table, the orthogonal design of conditions was carried out for tar-rich coal biodegradation by *B. licheniformis*. The parameters for orthogonal design are shown in Table 3.

### 2.5. Influences of Four Surfactants on Biodegradation Processes

#### 2.5.1. Biodegradation Extent

Under the optimal condition, 50 mL Rh (0, 400, 600, 800, 1000, 1200, 1400 mg/L) was added to the conical flasks before sterilization. After the operation, the biodegradation of coal was evaluated based on the method previously described. The pH value and surface tension of the supernatant liquid were determined after centrifugation. Surface and interface tension of the samples were determined using a Surface Tensiometer (Kruss 100 Tensionmeter, Kruss, Germany). The influences of Triton X-100, LAS, and DTAB on biodegradation of coal were studied using the same experimental method. A biodegradation experiment without surfactant was used as blank control group.

#### 2.5.2. *B. licheniformis*

A total of 50 mL surfactant solution (400 mg/L) was added to a 150 mL conical flask containing 20 mL of nutrient gravy agar medium. Then, the conical flask was wrapped with paper and sterilized in a vertical autoclave. After sterilization, the amplification culture of activated bacterium was inoculated and cultured in a constant-temperature shock incubator. The temperature was set at 30 °C, and the oscillation speed was 160 r/min.

The influences of four surfactants on *B. licheniformis* were evaluated using the following three methods:

(1) The influence of the surfactant on growth curve was studied by observing the OD_600_ and pH of *B. licheniformis* in liquid medium.

(2) The cell surface hydrophobicity (CSH) was measured using the bacterial adherence to hydrocarbons method [32]. The activated bacteria were transferred to a 100 mL nutritive gravy agar medium and cultured for 24 h. Then, the bacteria were collected by centrifugation and suspended in phosphate buffer (pH = 7.0). The optical density of the cell suspension was 0.5 (*A*_0_) at 600 nm. The CSH was calculated using Formula (2):(2)$$Q\left(\%\right)=\left(1-{A_{1}/A}_{0}\right)\times 100$$
where *Q* represents adsorption rate, and *A*_0_ and *A*_1_ represent the absorbance of the initial aqueous biomass and the aqueous biomass after mixing with dimethylbenzene at 600 nm, respectively.

(3) The permeability of the cell membrane was evaluated by the release of cytoplasmic b-galactosidase (ONP). A 1 mL cell suspension (OD_600_ = 0.5) was mixed with 18 mL surfactant solution, and 1 mL substrate o-nitrophenyl-b-D-galactopyranoside (ONPG, 30 mM) was added. After 0.5 h incubation at 30 °C, the culture sample was centrifuged at 4000 rpm for 5 min. Cultivate liquid without surfactant was used for blank control groups. Then, the absorbances of the supernatant liquid and the blank control group at 415 nm were measured. The yield of ONP was calculated using the following formula:(3)$$\eta_{ONP}=\frac{A_{415,i}-A_{415,0}}{\xi }$$
where *η_ONP_* is the yield of ONP (min^−1^·mL^−1^), *A*_415*,i*_ is the absorbance of the supernatant liquid at 415 nm, *A*_415,0_ represents the blank control groups, and ξ is the molar absorption coefficient of ONP (4.86 cm/mM).

#### 2.5.3. Coal

The wettability of surfactants on oxidized coal was compared by measuring the contact angles of four surfactants with oxidized coals. The contact angles of the coal were measured using JC2000 [33].

### 2.6. Analysis of Biodegradation Products

The degradation liquid products were extracted by toluene, dichloromethane, and ethyl acetate of different polarities. The extracts were analyzed using Agilent 7890A/5975 C gas chromatography–mass spectrometry. The GC-MS was equipped with a capillary column coated with HP-5MS (30 m × 0.25 mm × 0.25 µm). The parameters of gas chromatography were as follows: injector temperature, 280 °C; ion source temperature, 200 °C; temperature program: 60 °C, 180 °C at a rate of 6 °C/min, and held for 2 min; 300 °C at a rate of 10 °C/min, and held for 6 min.

FTIR characterizations of coal samples were performed using a Spectumn GX FTIR spectrometer at the transmission range from 4000 to 400 cm^−1^. The peak value of the spectrogram was fitted by OriginPro 2019b software. The heights, shapes, and peak areas were obtained by fitting the peaks [34].

## 3. Results and Discussions

### 3.1. Analysis of Biodegradation Conditions

#### 3.1.1. Analysis of Single-Factor Experiment

The influences of the amount of coal, the biodegradation time, and the inoculation volume on the A450 of the supernatant liquid are shown in Figure 2.

(1) Amount of coal

With the increase in the amount of coal, the A450 of the biodegradation liquid product increased initially and then leveled off. The results showed that low concentrations of coal depressed the efficiency of biodegradation, while the effect of high concentrations was not significant. The optimal amount of coal was 0.9 g (the concentration of coal was 18 g·L^−1^) (Figure 2a). This is because when the available bacteria nutrients were exhausted, they could not subsist with coal as the nutrient source. The amount of coal that could be degraded by bacteria was limited. Consequently, when the available nutrients for the bacteria were exhausted, the biodegradation effects of oxidized coals were no longer enhanced.

(2) Biodegradation time

The effect of the degradation time on the extent of coal biodegradation is presented in Figure 2b. With the increase in fermentation time, the extent of biodegradation of coal by bacteria before 9 days showed a gradual trend of increase, and there was no significant effect after 9 days. The optimal biodegradation time was 9 d. The reason for this may be that the survival of the strain depended on the medium. When the biodegradation time exceeded 9 d, the nutrients in the medium were exhausted, and the strain stopped metabolizing.

(3) Inoculation volume

The effect of the inoculation volume on the extent of coal biodegradation is presented in Figure 2c. The best degradation effect was obtained when the inoculation volume was 8 mL (the concentration of bacteria solution was 160 mL·L^−1^), while the further increase in concentration (>160 mL·L^−1^) had less of an influence on biodegradation (Figure 2c). This was because when the inoculation volume reached a certain amount, the degradable substances in coal were exhausted. The results showed that the changes in the inoculation volume influenced the extent of coal biodegradation.

#### 3.1.2. Analysis of Orthogonal Experiment

The orthogonal experiment results of the degradation of oxidized coal by *B. licheniformis* are listed in Table 4. The primary and secondary order of factors influencing the efficiency of coal biosolubilization was the amount of coal > biodegradation time > inoculation volume. The optimal biodegradation conditions were a coal amount of 0.7 g (the concentration of coal was 14 g·L^−1^), biodegradation time of 12 d, and inoculation volume of 8 mL (the concentration of bacteria solution was 160 mL·L^−1^). At this time, the degradation effect was the best, and the biodegradation rate was 43.5%.

### 3.2. Analysis of the Influences of Surfactants on Biodegradation

#### 3.2.1. Biodegradation Degree

The influences of the four surfactants on pH value, surface tension, and A450 of the supernatant liquid are shown in Figure 3. With the increase in the concentration of DTAB, the pH value, surface tension, and A450 of the supernatant liquid decreased. The reason for this might be that the acidic environment inhibited the growth of bacteria, and the bacteria could not degrade the oxidized coals through metabolism. However, with the increase in the concentration of Rh, Triton X-100, and LAS, each supernatant liquid always remained alkaline. The A450 of Rh, Triton X-100, and LAS were enhanced first and then weakened, and the optimal concentrations were 1200, 1000, and 1000 mg/L, respectively. The surface tensions of the supernatant liquid decreased first and then leveled off with the increase in Rh, Triton X-100, and LAS concentrations.

The biodegradation rates under the four surfactants are listed in Table 5. Under the actions of Rh, Triton X-100, LAS, and DTAB, the biodegradation rates were 51.6%, 59.8%, 54.3%, and 17.3%, respectively. Triton X-100 demonstrated the most significant improvement in biodegradation. The experimental results indicate that low concentrations of surfactants can promote coal biodegradation, but high concentrations have no obvious effect.

#### 3.2.2. Changes in *B. licheniformis*

(1) Growth curve

The OD_600_ and pH value changes of *B. licheniformis* in a liquid medium are shown in Figure 4. After adding Rh, Triton X-100, or LAS, the pH value of the medium remained alkaline, which was conducive to the growth of *B. licheniformis*, while the pH value in the medium was acidic in the presence of DTAB, which might inhibit the growth of strains.

When examining the OD_600_ observation of the growth curve, Rh, Triton X-100, and LAS promoted growth, while DTAB was the opposite. Che [35] reported that surfactants could destroy the structure of cell membranes by interfering with the growth cycle of bacteria, thereby reducing its activity. DTAB was toxic to *B. licheniformis*, resulting in a decrease in the biodegradation rate of coal.

(2) CSH

The bacterial cell surface is rich in hydrophobic and hydrophilic groups, and, among these, the adsorbed surfactant molecules might take the place of the hydrophilic moieties on the cell surface, with the hydrophobic tails extending to the environment. The changes in CSH of *B. licheniformis* under the actions of the four surfactants are shown in Figure 5. In the absence of surfactants, the CSH was 10.34%, while, in the presence of Rh, Triton X-100, LAS, and DTAB, the CSH was 13.15%, 18.00%, 14.29%, and 16.03%, respectively. The sequence in which Rh, Triton X-100, and LAS elevated bacterial CSH matched the sequence in which they enhanced biodegradation rates. Triton X-100 demonstrated the most significant improvement of CSH. Zhang [36] found that the increase in CHS stimulated the adsorption of pyrene and promoted the biodegradation of pyrene. The Rh, Triton X-100, and LAS might enhance the adsorption of macromolecules in coal on the cell surface and, consequently, guide their catabolic degradation.

(3) Cell membrane

The abilities of the four surfactants to permeate the cell membrane of *B. licheniformis* were evaluated by the release of cytoplasmic ONP. As shown in Figure 6, the yields of ONP under the actions of Rh, Triton X-100, LAS, DTAB, and surfactant-free control were 6.78%, 8.62%, 8.21%, 8.67%, and 6.20%, respectively. All surfactants showed a notable ability to promote cell permeability. The sequence in which Rh, Triton X-100, and LAS elevated cell permeability matched the sequence in which they enhanced biodegradation rates. Besides DTAB, Triton X-100 showed the most significant improvement in cell permeabilities. Van Hamme [37] found that the alteration of the membrane permeability was caused by the membrane absorption of surfactant molecules at relatively low surfactant concentrations. Li [10] reported that the increase in cell membrane permeability was conducive to the transmembrane transport rate of phenanthrene.

Thus, Rh, Triton X-100, and LAS enhanced the cell permeability, and the promotion of transmembrane transport helped cells to uptake and degrade coal molecules.

#### 3.2.3. Hydrophilicity of Coal

A low contact angle indicates hydrophilicity, while a high angle indicates more hydrophobic interaction. The measurement of the contact angle between oxidized coal and the four surfactant solutions is shown in Table 6.

All four surfactants improved the hydrophilicity of the oxidized coal, and the effect of the hydrophilicity of the surfactants on oxidized coal was in the following order: Triton X-100 (32.9°), LAS (38.2°), DTAB (40.9°), Rh (45.6°), and surfactant-free control (88.4°).

The increasing order of hydrophilicity under the action of Rh, Triton X-100, and LAS was also consistent with the increasing order of promoting coal biodegradation. Triton X-100 had the most significant improvement on oxidized coals. Li [38] found that surfactants enhanced the solubility of phenanthrene, which contributed to biodegradation. Consequently, all four surfactants augmented the hydrophilicity of oxidized coal, fostering an improved coal bioavailability.

Overall, Triton X-100 was ranked as the top performer in enhancing the CSH of bacteria and increasing the hydrophilicity of the oxidized coal. Additionally, it achieved second place in enhancing bacterial cell permeability. Among the tested agents, Triton X-100 exhibited the most significant enhancement of microbial degradation, establishing itself as the optimal choice for stimulating the biodegradation of tar-rich coal by *B. licheniformis*.

### 3.3. Characterization of Coal Biodegradation Products

The influence of Triton X-100 on the degradation product was studied with GC-MS and FTIR.

#### 3.3.1. GC-MS Analysis of Degradation Liquid Products

The changes in degradation liquid products of coal biodegradation before and after the action of Triton X-100 were studied using GC-MS.

The GC-MS spectra of toluene extracts from degradation liquid products before and after the action of Triton X-100 showed similar trends (Figure 7), with 14 identical compounds. The main components of the toluene extract liquid are shown in Figure 8, and details are listed in Table 7. The toluene extract liquid from the control group mainly contained alkanes (77.07%), aromatic hydrocarbons (9.79%), esters (7.87%), and amines (5.27%), totaling 34 compounds. After adding Triton X-100, the number of identified degradation compounds was reduced to 18, among which the content of alkanes decreased to 58.88%, aromatics decreased to 1.93%, esters increased to 39.19%, and amines disappeared. Under the action of Triton X-100, hexadecanoic acid, 2-hydroxy-1-(hydroxymethyl) ethyl ester increased from 2.11% to 20.67%, and nonadecane (11.19%) disappeared. Among the four newly produced substances, the content of hexadecane was 9.90% and the content of octadecanoic acid, 2,3-dihydroxypropyl ester was 16.37%.

The GC-MS of the dichloromethane extract is shown in Figure 9; the trend of the liquid products before and after the action of Triton X-100 was similar, with 14 compounds being identical. The main components of the dichloromethane extract liquid are shown in Figure 8, and details are listed in Table 8. The dichloromethane-extracted liquid from the control group contained 26 compounds, mainly including alkanes, comprising 80.91%; aromatics, comprising 6.56%; esters, comprising 5.77%; amines, comprising 5.09%; and carboxylic acids, comprising 1.67%. With the surfactant treatment, the number of compounds in the dichloromethane extract liquid reduced to 20, only retaining alkanes (67.48%) and esters (32.52%), while carboxylic acids, amines, and aromatics disappeared. Meanwhile, hexadecane increased from 5.34% to 15.10%, hexadecanoic acid, 2-hydroxy-1-(hydroxymethyl) ethyl ester increased from 3.57% to 17.50%, and octadecanoic acid, 3-dihydroxypropyl ester increased from 2.20% to 13.11%.

The GC-MS of ethyl acetate extract is shown in Figure 10; the trend of liquid products before and after the action of Triton X-100 was similar, with 1 compound being identical. The main components of the toluene extract liquid are shown in Figure 8, and the details are listed in Table 9. The main components of ethyl acetate extract liquid are shown in Table 7 and Table 9. The ethyl-acetate-extract liquid from the control group contained three compounds, mainly esters (86.03%), amides (10.46%), and arenes (3.50%). With surfactant treatment, the number of identified degradation compounds in ethyl acetate of the liquid product increased to nine, the esters (86.40%) and arenes (3.87%) had little change, and carboxylic acids (9.73%) appeared while amine disappeared. Among the six newly produced substances, the content of hexadecanoic acid, 2-hydroxy-1-(hydroxymethyl) ethyl ester comprised 13.58%, and octadecanoic acid, 2-hydroxy-1-(hydroxymethyl) ethyl ester comprised 10.69%.

In conclusion, after the action of Triton X-100, the number of identified degradation compounds in the toluene and dichloromethane extract of the liquid product decreased by 16 and 6, respectively, but the number in the ethyl acetate extract increased by 6. At the same time, the contents of alkanes in the extracts of toluene and dichloromethane decreased, and lipids increased while the ethyl acetate extract changed little. The degraded liquid products were mainly aromatic hydrocarbons, long-chain alkanes, aldehydes, ketones, and ether esters, consistent with Akimbekov’s report [30]. The results also showed that adding suitable surfactants could promote the enrichment of degradation products. In liquid products, alkanes are commonly used as industrial fuels; octadecanoic acid, 2,3-dihydroxypropyl ester is widely used as emulsifiers in the food, medicine, cosmetics, and detergent industries; and dibutyl phthalate is used as a plasticizer for acetate fiber, polyvinyl chloride, etc.

#### 3.3.2. FTIR Analysis of the Coal Samples

The FTIR spectra of TRC, RC, oxidized coal, and raw coal are shown in Figure 11, and the peak shape trends of the four coal samples are similar. The absorption bands of products were at 3420 cm^−1^ and 1600 cm^−1^, which were the stretching vibration peaks of the carboxyl group and aromatic ring, respectively. After nitric acid treatment, new absorption bands appeared in oxidized coal at 1716 cm^−1^, 1540 cm^−1^, and 1320 cm^−1^, which were the characteristic peaks of the carboxyl group, nitro group, and hydroxyl group, respectively. These changes indicated that nitric acid pretreatment increased the number of carbon-based functional groups in oxidized coals and introduced -NH_2,_ meaning that the raw coal was oxidized by nitric acid [17]. The absorption bands of residual coal from the control group disappeared at 1540 cm^−1^, indicating that the bacteria degraded the fragments containing C=O and N=O functional groups from the macromolecular structure of coal. After the action of Triton X-100, new absorption bands appeared at 2910 cm^−1^ and 2850 cm^−1^, which were the absorption peaks of aliphatic methyl and methylene. The results showed that Triton X-100 promoted *B. licheniformis* to degrade the aromatic ring in coal, increasing the aliphatic substances. The absorption peaks at 753 cm^−1^ were the characteristic peaks of aromatic vibration [39]. Compared with oxidized coal, there was a new absorption band of RC and TRC, which might be due to the depolymerization of polycyclic aromatic hydrocarbons in it.

(1) Changes in aromatic structures

The aromatic structure parameters of four coal samples are listed in Table 10. After nitric acid oxidation, *H_ar_*/*H* of raw coal decreased from 0.64 to 0.33. The reason for this may be that the aromatic ring in tar-rich coal was destroyed by nitric acid. Zhang et al. [40] found that oxygen atoms mainly attacked the aromatic ring and entered the coal structure by forming oxides. Therefore, the fatty acids in oxidized coal decreased while the aliphatic groups increased.

After degradation by *B. licheniformis*, the *H_ar_*/*H* of oxidized coal decreased from 0.33 to 0.27, indicating that the aromatic ring in oxidized coal was degraded by *B. licheniformis*. The aromatic ring might be damaged by amylase, protease, chitinase, and alkaline protease secreted by *B. licheniformis* [19]. A large number of aromatic ring structures were also observed in the GC-MS analysis. Furthermore, compared with RC, the *H_ar_*/*H* of TRC was lower, which indicated that the Triton X-100 enhanced the degradation of aromatic structures of coal oxidized by *B. licheniformis*.

(2) Changes of aliphatic structures

As shown in Table 10, after nitric acid oxidation, the *A*(CH_2_)/*A*(CH_3_) of raw coal increased from 1.74 to 3.15, which indicated that nitric acid oxidation destroyed the aromatic ring in raw coal and increased the side chain alkyl of oxidized coal. Compared with oxidized coal, the *A*(CH_2_)/*A*(CH_3_) of RC decreased from 3.15 to 2.46, indicating that side chain alkyls in oxidized coal were degraded by *B. licheniformis*. A large number of aliphatic structures were also observed in the GC-MS analysis. Moreover, compared with RC, the *A*(CH_2_)/*A*(CH_3_) of TRC was lower, which showed that the Triton X-100 enhanced the degradation of aliphatic structures of coal oxidized by *B. licheniformis*.

The changes in the chemical composition of degradation products indicated that surfactants were an effective way to modulate the process of coal biodegradation. Therefore, adding suitable surfactants could promote biodegradation and enrich the degradation products. However, due to the complexity of the composition of coal, the industrial application of coal samples, strains, and surfactants in this paper needs to be further investigated.

### 3.4. Mechanism of Coal Biodegradation under the Action of Triton X-100

The possible process by which Triton X-100 promotes the biodegradation of coal is shown in Figure 12. On the one hand, Triton X-100 could reduce the interfacial tension between oxidized coal, macromolecules, and water and promote the diffusion of macromolecules from the coal to the water phase. This transfer of macromolecules increased the possibility of microbial contact with macromolecules and promoted the microbial degradation of oxidized coals. On the other hand, surfactants also increased the hydrophobicity of the cell surface, making it easier for macromolecular substances in coal to be adsorbed by the surface of the bacteria [41]. In addition, the cell membrane of the strain was amphiphilic, which was similar to the structure of the surfactant. Surfactant molecules changed the permeability of the cell membrane by inserting into the cell membrane so as to promote the macromolecules in coal to enter the bacterium, and the enzyme secreted by the bacterium was transferred out of the cell membrane [42,43].

The possible pathway by which *B. licheniformis* degrades coal macromolecules was hypothesized based on the classical coal macromolecules, as shown in Figure 12 [44]. The main characteristic of tar-rich coal is its hydrogen-rich structure, mainly composed of the side chain and bridge bond with adipose structure in coal. The bridge bonds in coal were mainly adipose structures (-CH_2_-, -CH_2_-CH_2_-, -CH_2_-O-, etc.), and the side chains were mainly composed of alkyl and heteroatomic groups [1].

In the initial stage of coal biodegradation, the alkaline substances secreted by *B. licheniformis* act as solvents, effectively penetrating into the coal matrix and, to some extent, depolymerizing tar-rich coal [45]. Subsequently, the widening of the internal space of coal macromolecules could allow more enzymes to enter the coal matrix, and the presence of the surfactant Triton X-100 could enhance enzyme adsorption on coal by changing the charge and hydrophilicity of the coal surface [46]. The contact of enzymes with tar-rich coal provided more opportunities for coal biodegradation. Moreover, extracellular enzymes may directly oxidize aromatic compounds, open aromatic rings, and cleave the side chains of aromatic rings by introducing more C-O groups into the coal structure to substitute H in the aromatic ring [47]. According to GC-MS and FTIR analysis, the results showed that, during the degradation of oxidized coals by *B. licheniformis*, the aromatic ring and side chain alkyl in oxidized coals were broken and formed simple small molecular substances, such as alkanes, carboxylic acids, esters, and aromatic compounds. Jiang [29] also reported that *Bacillus* sp. *Y7* degraded coal into small molecules through hydroxylation, demethylation, and hydrolysis.

## 4. Conclusions

The biodegradation rates under the actions of Triton X-100, LAS, Rh, DTAB, and the control group were 59.8%, 54.3%, 51.6%, 17.3%, and 43.5%, respectively. Triton X-100 demonstrated the most significant improvement in the biodegradation process when the conditions were as follows: amount of coal, 14 g·L^−1^; Triton X-100, 1000 mg·L^−1^; biodegradation time, 12 d; inoculation volume, 160 mL·L^−1^.

The growth curve of *B. licheniformis* showed that Rh, Triton X-100, and LAS could promote growth, while DTAB was toxic to the bacterium. The cell permeability of the bacterium was improved by each surfactant, which was conducive to the transmembrane transport of macromolecules in coal into the bacterium, and the order was DTAB > Triton X-100 > LAS > Rh > control group. The CSH of the bacterium in the presence of Triton X-100, DTAB, LAS, Rh, and the control group was 18.00%, 16.03%, 14.29%, 13.15%, and 10.34%, respectively. The contact angles between coal surfaces and surfactant solutions obeyed the following order: Triton X-100 (32.9°), LAS (38.2°), DTAB (40.9°), Rh (45.6°), and surfactant-free control (88.4°). Triton X-100 ranked first in improving the CSH of bacteria in the four surfactants and the hydrophilicity of oxidized coal and second in improving the microbial cell permeability. Hence, Triton X-100 was the optimal choice for facilitating the biodegradation of tar-rich coal by *B. licheniformis*.

The GC-MS showed that the degraded liquid products were mainly aromatic hydrocarbons, long-chain alkanes, aldehydes, ketones, and ether esters. After Triton X-100 treatment, the number of identified degradation compounds in the toluene extract of the liquid product decreased by 16 compared to the control group; the number of compounds extracted from dichloromethane decreased by 6, and those in the ethyl acetate extract increased by 6. At the same time, the contents of alkanes in the extracts of toluene and dichloromethane decreased, and lipids increased, but the ethyl acetate extract changed little. The FTIR analysis suggested that, under the action of surfactants, compared to oxidized coal, the *H_ar_*/*H* and *A*(CH_2_)/*A*(CH_3_) of the remaining coal decreased by 0.07 and 1.38, respectively, indicating that Triton X-100 enhanced the degradation of aromatic and aliphatic structures of oxidized coal. Therefore, adding suitable surfactants could promote biodegradation and enrich the degradation products.

In summary, surfactants significantly modify surface activity on coal samples, bacteria, and degradation products and are an effective method that can be used to regulate the biodegradation process of coal.

## Figures and Tables

**Figure 1 microorganisms-11-02397-f001:**
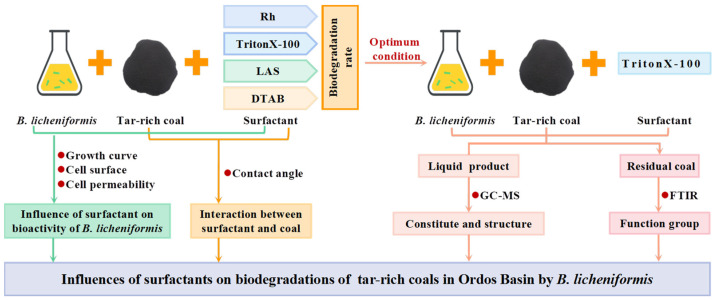
Degradation experimental process.

**Figure 2 microorganisms-11-02397-f002:**
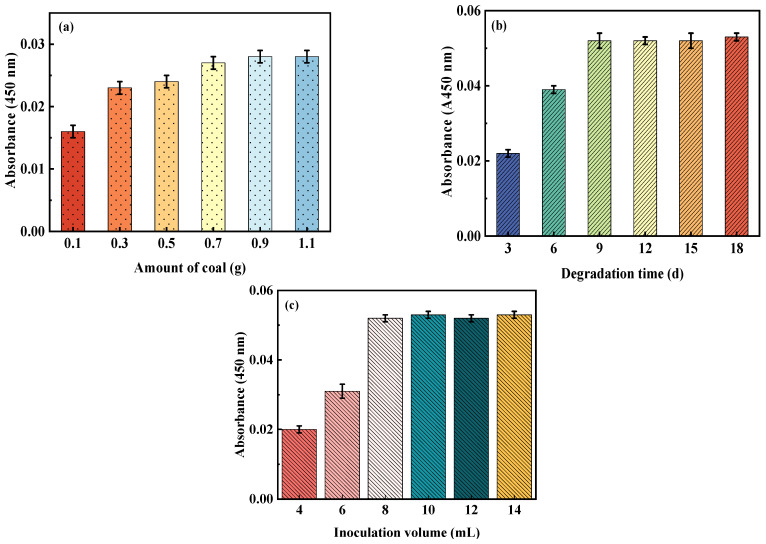
Influences of three factors on A450 of the supernatant liquid. (**a**) Amount of coal; (**b**) biodegradation time; (**c**) inoculation volume.

**Figure 3 microorganisms-11-02397-f003:**
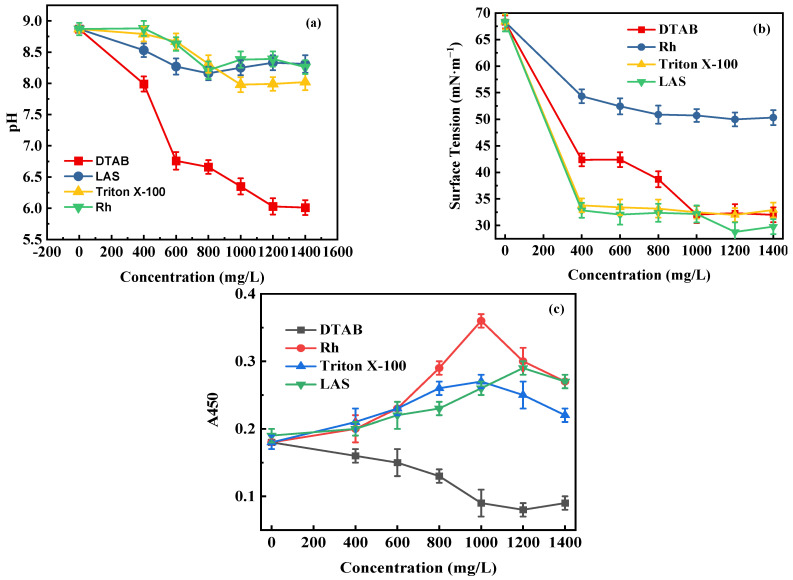
Influences of concentrations of four surfactants on (**a**) pH value; (**b**) surface tension; (**c**) A450 of the supernatant liquid.

**Figure 4 microorganisms-11-02397-f004:**
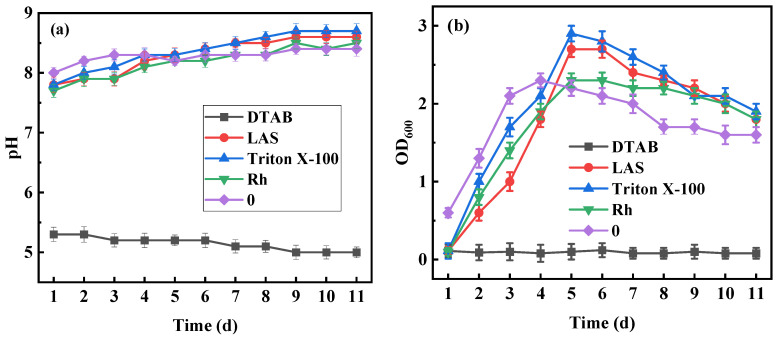
Influences of four surfactants on the growth processes of *B. licheniformis*. (**a**) Growth environment; (**b**) growth curve.

**Figure 5 microorganisms-11-02397-f005:**
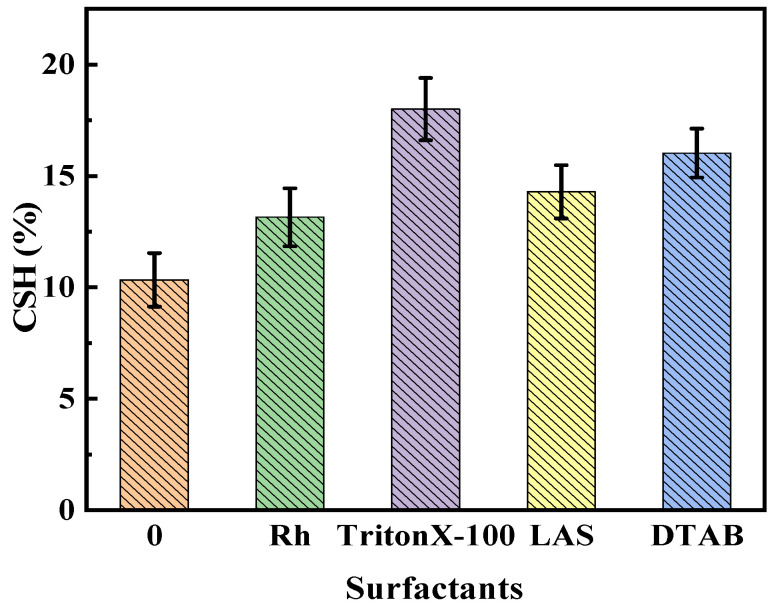
Influences of four surfactants on the CSH of *B. licheniformis*.

**Figure 6 microorganisms-11-02397-f006:**
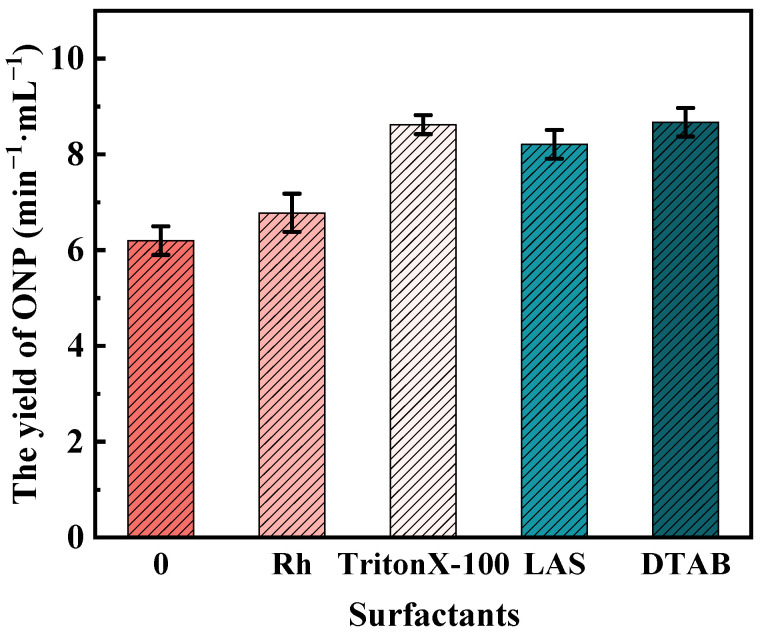
Influences of four surfactants on the permeability of the cell membrane of *B. licheniformis*.

**Figure 7 microorganisms-11-02397-f007:**
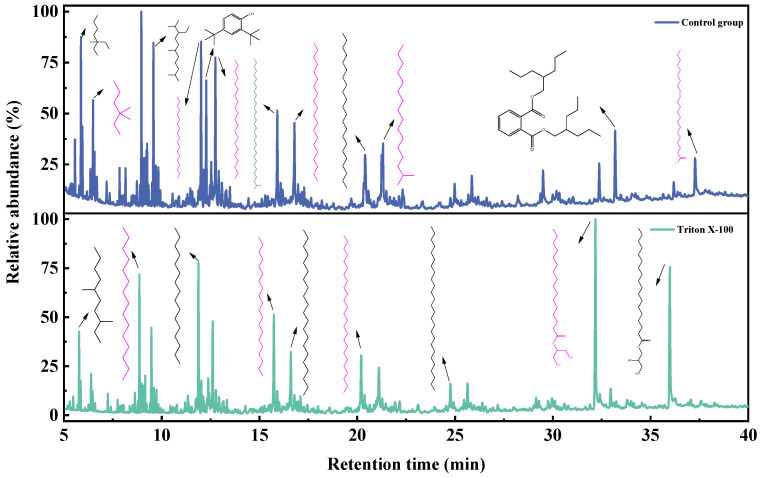
GC-MS spectra of toluene extracted from degradation liquid products before and after the actions of Triton X-100.

**Figure 8 microorganisms-11-02397-f008:**
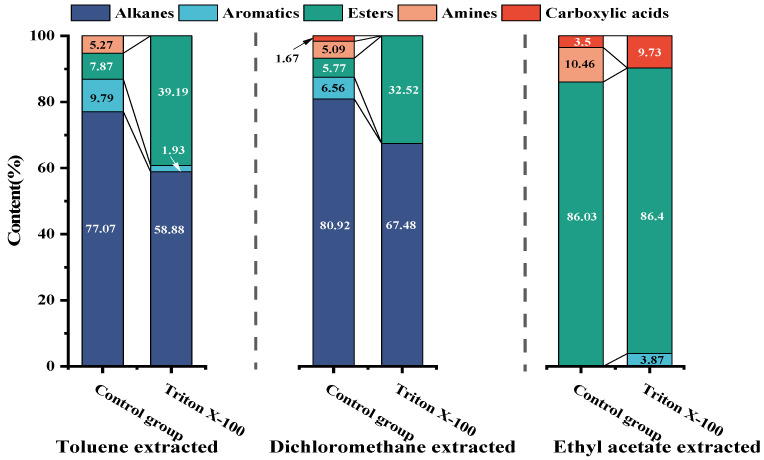
The compositions of extracts from the degradation liquid products before and after the actions of Triton X-100.

**Figure 9 microorganisms-11-02397-f009:**
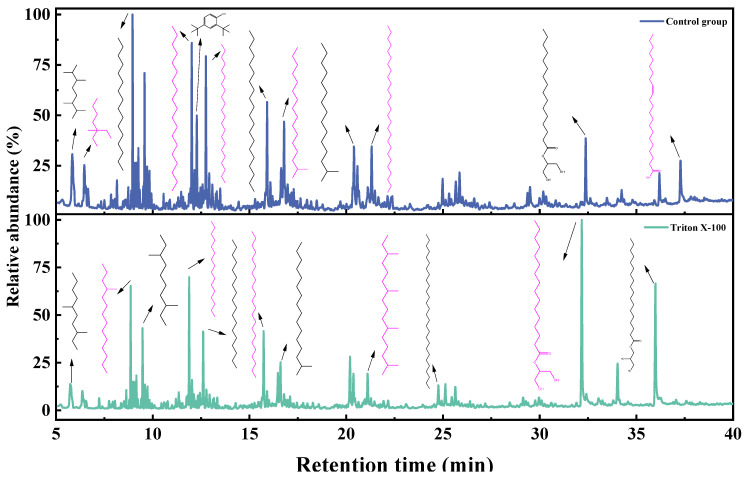
GC-MS spectra of dichloromethane extracted from the degradation liquid products before and after the actions of Triton X-100.

**Figure 10 microorganisms-11-02397-f010:**
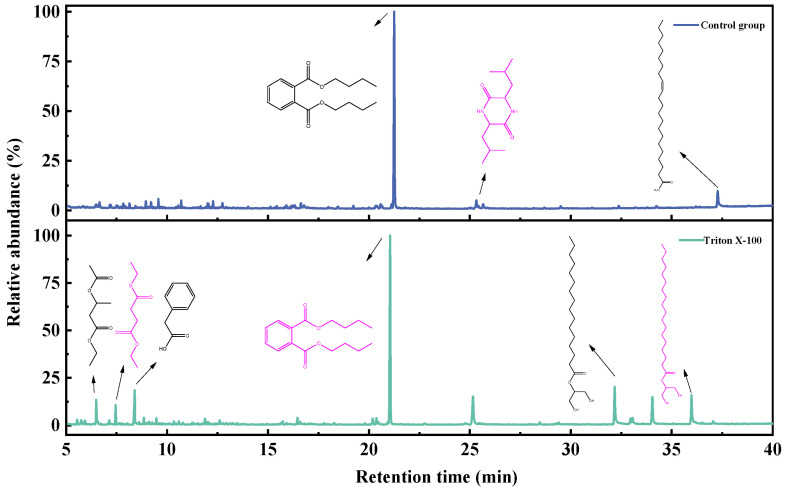
GC-MS spectra of ethyl acetate extracted from the degradation liquid products before and after the actions of Triton X-100.

**Figure 11 microorganisms-11-02397-f011:**
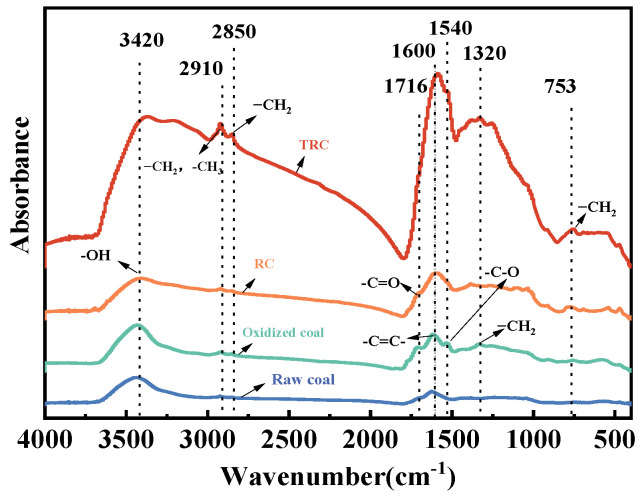
FTIR spectra of TRC, RC, oxidized coal, and raw coal. The residual coal degraded by *B. licheniformis* is referred to as RC, and, under the action of Triton X-100, the residual coal is referred to as TRC.

**Figure 12 microorganisms-11-02397-f012:**
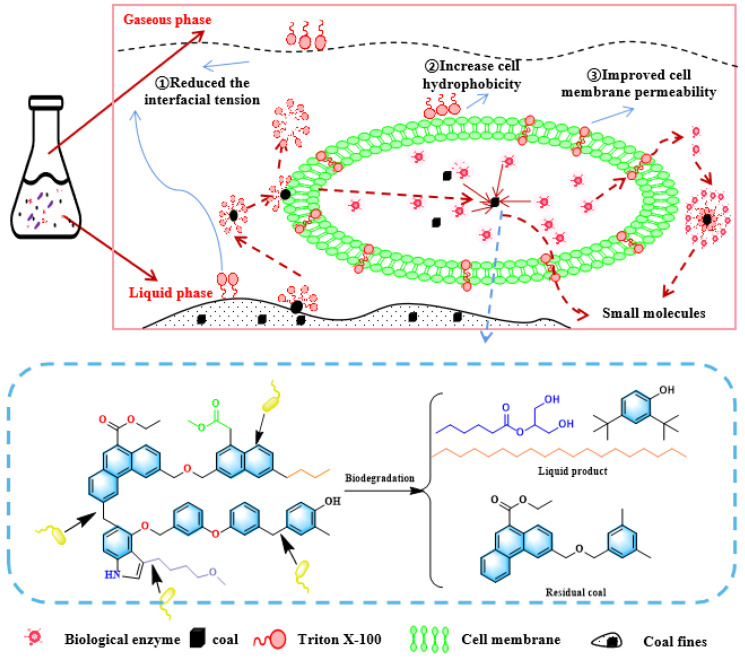
The process of coal biodegradation under Triton X-100.

**Table 1 microorganisms-11-02397-t001:** Proximate and ultimate analysis of the coal samples.

Coal	Proximate Analysis/%				Ultimate Analysis/%				
	*M* _ad_	*A* _ad_	*V* _ad_	FC *	C_ad_	H_ad_	N_ad_	S_t,ad_	O *
Raw coal	7.10	9.32	28.52	55.06	60.24	3.18	4.50	0.12	31.96
Oxidized coal	2.72	3.08	40.04	54.16	58.34	3.66	4.25	0	33.75

* expressed as difference subtraction method.

**Table 2 microorganisms-11-02397-t002:** The structures of surfactants.

Name	Abbr.	Classification	Structure
Rhamnolipid	Rh	Biological surfactant	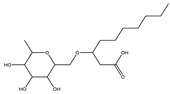
Polyethylene glycol octylphenyl ether	Triton X-100	Nonionic surfactant	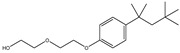
Straight chain alkyl benzene sulfonate	LAS	Anionic surfactant	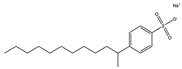
Dodecyltrimethylammonium bromide	DTAB	Cationic surfactant	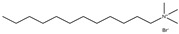

**Table 3 microorganisms-11-02397-t003:** The design of single-factor experiments and orthogonal experiments.

Experiment	No.	A-Amount of Coal (g)	B-Biodegradation Time (d)	C-Inoculation Volume (mL)
Single-factor experiment	1	0.1	3	4
	2	0.3	6	6
	3	0.5	9	8
	4	0.7	12	10
	5	0.9	15	12
	6	1.1	18	14
Orthogonal experiment	1	0.7	6	6
	2	0.9	9	8
	3	1.1	12	10

**Table 4 microorganisms-11-02397-t004:** The results of the orthogonal experiment.

No.	A	B	C	A450
1	1	1	1	0.021
2	2	2	2	0.023
3	3	3	3	0.022
4	3	1	2	0.023
5	1	2	3	0.031
6	2	3	1	0.025
7	2	1	3	0.024
8	3	2	1	0.026
9	1	3	2	0.032
*K* _1j_	0.084	0.068	0.072	
*K* _2j_	0.072	0.080	0.078	
*K* _3j_	0.071	0.079	0.077	
*R*	0.013	0.012	0.006	

**Table 5 microorganisms-11-02397-t005:** Degradation rates under the actions of different surfactants.

Degradation Types	Biodegradation Rate
Oxidized coal + *B. licheniformis*	43.5% ± 2.3%
Oxidized coal + *B. licheniformis* + Rh	51.6% ± 2.6%
Oxidized coal + *B. licheniformis* + Triton X-100	** 59.8% ± 3.2% **
Oxidized coal + *B. licheniformis* + LAS	54.3% ± 3.6%
Oxidized coal + *B. licheniformis* + DTAB	17.3% ± 1.4%

**Table 6 microorganisms-11-02397-t006:** Influences of four surfactants on contact angles of coal samples.

No.	Rh	Triton X-100	LAS	DTAB	Surfactant-Free Control
Contact angle (°)	45.6 ± 1.2	32.9 ± 1.4	38.2 ± 1.6	40.9 ± 1.3	88.4 ± 1.7

**Table 7 microorganisms-11-02397-t007:** Main compounds in the toluene extracted from the degradation liquid products before and after the actions of Triton X-100.

Control Group	Triton X-100
No.	Retention Time(min)	Content(%)	Compounds	Retention Time(min)	Content(%)	Compounds
1	5.962	2.27	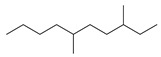	5.783	4.28%	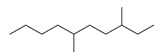
2	7.846	1.24	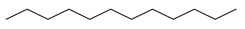	6.388	1.40%	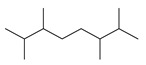
3	9.158	2.75	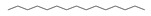	8.856	9.55%	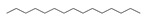
4	12.535	1.50	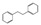	9.047	2.25%	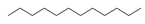
5	12.759	8.16	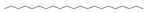	9.596	2.61%	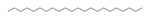
6	12.927	2.64	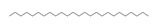	11.884	7.15%	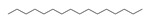
7	13.084	1.62	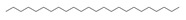	12.030	2.77%	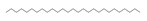
8	16.090	2.13	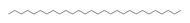	12.120	6.53%	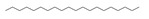
9	16.797	6.16	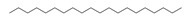	12.378	1.39%	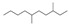
10	20.409	4.29	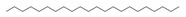	12.602	12.37%	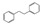
11	22.147	1.43	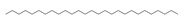	12.771	12.40%	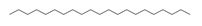
12	24.985	2.34	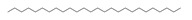	16.786	2.87%	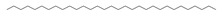
13	32.377	2.11	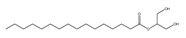	17.100	1.94%	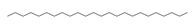
14	33.195	3.43	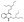	23.123	1.39%	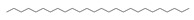
15	5.569	1.37	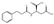	24.772	2.80%	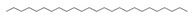
16	5.872	3.35	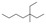	32.186	14.92%	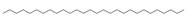
17	6.399	1.45	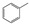	32.960	1.55%	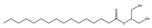
18	6.489	2.63	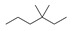	35.989	11.82%	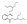
19	9.080	1.25	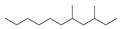	-	-	-
20	9.259	1.34	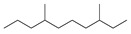	-	-	-
21	9.585	4.02	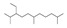	-	-	-
22	9.708	1.64	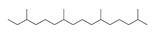	-	-	-
23	12.019	11.19	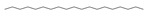	-	-	-
24	12.164	2.44	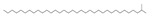	-	-	-
25	12.288	4.07	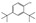	-	-	-
26	15.911	4.33	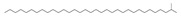	-	-	-
27	16.976	2.68	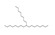	-	-	-
28	17.290	1.33	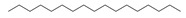	-	-	-
29	21.250	2.33	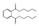	-	-	-
30	21.317	3.94	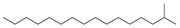	-	-	-
31	22.327	1.41	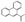	-	-	-
32	29.516	2.54	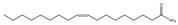	-	-	-
33	30.201	1.90	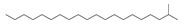	-	-	-
34	37.289	2.73	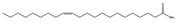	-	-	-

**Table 8 microorganisms-11-02397-t008:** Main compounds in the dichloromethane extracted from the degradation liquid products before and after the actions of Triton X-100.

Control Group	Triton X-100
No.	Retention Time(min)	Content(%)	Compounds	Retention Time(min)	Content(%)	Compounds
1	5.839	5.78%	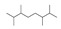	5.738	4.88%	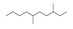
2	6.467	3.08%	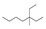	6.366	2.29%	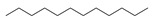
3	8.968	5.34%	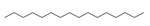	8.856	15.10%	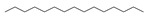
4	9.08	2.65%	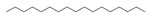	8.968	1.40%	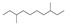
5	9.159	2.34%	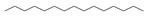	9.159	3.40%	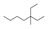
6	9.26	1.54%	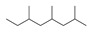	9.473	3.51%	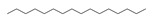
7	9.585	10.28%	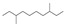	9.596	1.66%	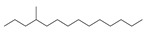
8	9.708	1.82%	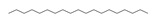	12.03	9.30%	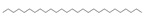
9	9.843	2.79%	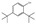	12.434	3.26%	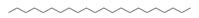
10	11.469	6.73%	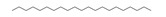	12.602	4.96%	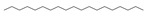
11	12.019	6.38%	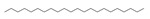	12.771	1.49%	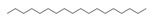
12	12.288	3.63%	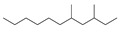	15.9	1.16%	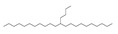
13	12.58	8.93%	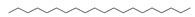	16.472	3.01%	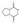
14	12.928	2.82%	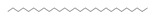	16.607	3.99%	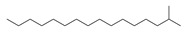
15	16.64	2.93%	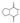	16.786	3.34%	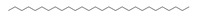
16	16.797	7.02%	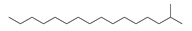	21.104	3.09%	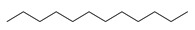
17	17.167	2.16%	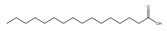	25.647	1.65%	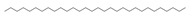
18	20.409	3.98%	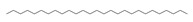	32.175	17.50%	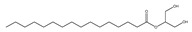
19	21.127	1.67%	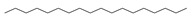	32.411	1.91%	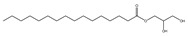
20	21.317	4.00%	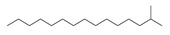	35.989	13.11%	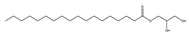
21	24.985	1.78%	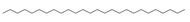	-	-	-
22	25.86	1.50%	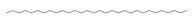	-	-	-
23	29.506	1.77%	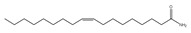	-	-	-
24	32.388	3.57%	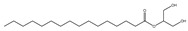	-	-	-
25	36.202	2.20%	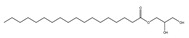	-	-	-
26	37.29	3.32%	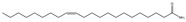	-	-	-

**Table 9 microorganisms-11-02397-t009:** Main compounds in the ethyl acetate extracted from the degradation liquid products before and after the actions of Triton X-100.

Control Group	Triton X-100
No.	Retention Time(min)	Content(%)	Compounds	Retention Time(min)	Content(%)	Compounds
1	21.250	86.03	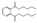	6.489	5.47	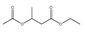
2	25.322	3.50	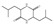	7.454	3.36	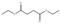
3	37.279	10.46	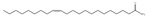	8.396	9.73	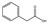
4	-	-	-	16.461	1.96	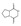
5	-	-	-	20.185	1.92	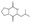
6	-	-	-	21.048	51.64	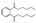
7	-	-	-	32.175	13.58	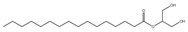
8	-	-	-	32.96	1.66	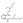
9	-	-	-	35.978	10.69	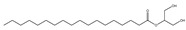

**Table 10 microorganisms-11-02397-t010:** Structural parameters of TRC, RC, oxidized coal, and raw coal.

Coal	*H_ar_*/*H*	*A*(CH_2_)/*A*(CH_3_)
	*A*_700–900_*/*(*A*_2800–3000_ + *A*_700–900_)	*A* _2915–2940_ */A* _2950–2975_
Raw coal	0.64	1.74
Oxidized coal	0.33	3.15
RC	0.27	2.46
TRC	0.20	1.08

## Data Availability

The data that support the findings of this study are available from the corresponding author upon reasonable request.

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
