# Peer review of "Influences of Four Kinds of Surfactants on Biodegradations of Tar-Rich Coal in the Ordos Basin by Bacillus bicheniformis"

_microorganisms, 2023, doi:10.3390/microorganisms11102397_

Round 1

Reviewer 1 Report

The article is devoted to the study of the effect of four different surfactants on the tar-rich Coal biodegradation by Bacillus bicheniformis. Undoubtedly, the study of the utilization of tar-rich coals using biological (environmentally safer) methods is relevant. The authors used a wide range of research methods, including modern ones, which made it possible to obtain new interesting data.

A few points:

1. Page 5, line 182. Need to correct 410 nm to 415 nm.

2. Section 2.6. Analysis of biodegradation products. Information on gas chromatography-mass spectrometry and FTIR conditions should be expanded. It is necessary to describe the method of sample preparation in details (what kind of extraction, volumes of samples and solvents, and other conditions) and the method of GCMS analysis (column used, column flow, temperatures of injector, interface, ion source, the program of temperature gradient, scanned m/z interval). Did you use the internal standard for calculation? Describe the method of quantification, please.

3. There is no section on statistical analysis of results.

4. Section 3 Results should be replaced by Results and discussions

5. In Fig.2 and further in the text, it is more correct to replace the phrase «Inoculation size» with «Inoculation volume».

6. Please indicate on what day of strain growth the data on the effect of the amount of coal and the volume of inoculation on the density of the suspension were presented (Fig2a and 2c).

7. In Fig. 4 and Fig.2 should be indicated error bar.

8. Fig. 3 should be moved to section 3.2.1.

9. Table 4 should be moved to section 3.1.2.

10. Table 5 should be moved to section 3.2.1.

11. From Line 334 and further. “…34 types of compounds”.  I would suggest to write “34 compounds” as they are individual compounds, but not a type.

12. Table 7 and 8. I can mention some flaws in the compound identification.

The retention time is an important characteristic for the compound identification. Some rules are existed.

-          The longer the aliphatic chain the shorter the retention times. Therefore, it is necessary to check the identification for compounds NN 5-12 (Table 7) and compounds NN 2,3,5,6,10,11,12 (Table 8).

-          A particular compound should have the same retention time in different samples. How can you explain a significant shift of Rt for compounds NN 2,5,6,7,8,9,10,11 (Table 7) and compounds NN 1,2,5,6,7,10,11 (Table 8).

13. Line 399 “The residual coal degraded by B. licheniformis is referred to as RC and under the  action of Triton X-100 the residual coal is referred to as TRC”. Please, move this sentence to the caption of the picture.

14. Line 489 and further:  “…the types of toluene extract in the degradation liquid products decreased by 16 compared to control group… ”  change to “…the amount of  the identified degradation compounds in  toluene extract of the liquid  product decreased by 16 compared to control group… ”

It is highly recommended to check the language as some sentences may be difficult to understand.

Reviewer 2 Report

This Manuscript ID: microorganisms-2614567 entitled  "Influences of Four Kinds of Surfactants on Biodegradations of Tar-rich Coal in Ordos Basin by Bacillus Bicheniformis" is  presents a comprehensive study on the biodegradation of tar-rich coal  using Bacillus licheniformis

The manuscript is generally well-written and offers interesting insights, but it does have some areas that require clarification and further elaboration.

 Specific Comments:

Introduction:

The introduction well enough but in need to be elaborated why the specific surfactants were chosen for the study.

Methods:

Please specify the conditions under which the biodegradation rates were measured. What was the duration of the study, and were any controls run alongside?

Explain why these specific surfactants were chosen for your research.

Results:

Elaborate more on analysis of biodegradation conditions

Elaborate more on how you determined the mechanisms of biodegradation.

Were there any molecular or genomic analyses possible on B. licheniformis?

The paper would be strengthened by a more in-depth comparative analysis between the surfactants.

GC-MS and FTIR analysis sections are informative. Please state, how do the changes in chemical composition relate to the potential applications or limitations of this biodegradation process?

Discuss limitations and provide discussion on feasibility  

Conclusions:

The conclusion summarizes the key findings well but could also mention potential implications for industrial applications and future research directions. Remove numbering of paragraphs.

Minor Revision
